# Effect of In Vitro Addition of Melatonin and Glutathione on Seminal Parameters of Rams in Diluted Semen and after Thawing

**DOI:** 10.3390/vetsci10070446

**Published:** 2023-07-08

**Authors:** Cristiana Carriço, João Pedro Barbas, Jorge Pimenta, João Simões

**Affiliations:** 1Department of Veterinary Sciences, Animal and Veterinary Research Centre (CECAV), Associate Laboratory for Animal and Veterinary Sciences (AL4AnimalS), School of Agricultural and Veterinary Sciences, University of Trás-os-Montes and Alto Douro (UTAD), 5000-801 Vila Real, Portugal; cristianaplcarrico@gmail.com; 2Department of Biotechnology and Genetic Resources of Instituto Nacional de Investigação Agrária e Veterinária (INIAV), Quinta da Fonte Boa, 2005-048 Vale de Santarém, Portugal; pedro.barbas@iniav.pt (J.P.B.); jorge.pimenta@iniav.com (J.P.); 3CIISA—AL4AnimalS, Faculty of Veterinary Medicine, University of Lisbon, 1649-004 Lisboa, Portugal

**Keywords:** sperm cryopreservation, semen quality, antioxidants, sperm motility, rams

## Abstract

**Simple Summary:**

The cryopreservation of semen has numerous advantages but decreases sperm quality, compromising its viability and functionality. Oxidative stress is considered the main cause of this decrease, so using antioxidants such as glutathione (GSH) and melatonin (MLT) may be a way to overcome this problem. We aimed to evaluate whether adding 1 mM MLT, 5 mM GSH, or both improves ram semen quality after thawing. We observed that, in fresh semen, the separate addition of MLT and GSH had a negative effect on sperm motility parameters. In thawed semen, MLT increased progressive sperm motility and combined with MLT, attenuated the negative effects of GSH. This study provides valuable information for selecting antioxidants or possible associations to be added to ram freezing-semen extenders.

**Abstract:**

The present study aimed to evaluate the effects of melatonin (MLT), glutathione (GSH), and their combination on ram semen quality after thawing. During eight weekly sessions, semen from three Merino rams was pooled, diluted with an egg-yolk-based semen extender, and divided into four groups: control, 1 mM MLT, 5 mM GSH, and 1 mM MLT + 5 mM GSH. Diluted semen was evaluated before and after the freezing process. The supplementation of diluted semen with GSH at 5 mM had a deleterious effect on total motility progressive (TPM), curvilinear velocity (VCL), straight-line velocity (VSL), average-path velocity (VAP), linearity (LIN), and straightness (STR) and increased slow spermatozoa (%). MLT at 1 mM also had a negative effect on TPM, VSL, and STR in diluted semen. In thawed semen, 1 mM MLT increased the TPM compared with the control group. VSL was lower in the 5 mM GSH group than in the 1 mM MLT group. Additionally, the combination of both antioxidants attenuated the negative effect of 5 mM GSH on TPM, VSL, and BCF. These results indicate that 5 mM GSH impairs or does not improve sperm kinetic parameters in either diluted or thawed semen. They also suggest that MLT combined with GSH plays a protective role against these effects.

## 1. Introduction

Semen cryopreservation preserves semen quality for future use [1,2]. However, there is a loss of 40–50% in sperm viability after thawing [3], and optimization protocols are needed. Freezing and thawing procedures expose spermatozoa (SPZ) to adverse conditions and oxidative stress caused by the excessive formation of reactive oxygen species (ROS) [4,5]. At the physiological level, ROS are important for normal SPZ function; however, in situations of oxidative stress, high levels of these compounds compromise sperm viability, causing morphological and functional damage through the oxidation of lipids, proteins, sugars, and DNA [5,6,7]. Procedures such as freezing, thawing, centrifugation, and incubation are thought to produce significant amounts of ROS, with a negative impact on sperm quality parameters and reproductive competence [8]. It is believed that hydrogen peroxide (H_2_O_2_) is the main ROS responsible for these forms of damage, resulting in decreased SPZ motility and viability and changes that hinder sperm fusion with the oocyte during fertilization [7].

To protect SPZ from oxidative stress, there is an intrinsic antioxidant system present in seminal plasma [9] and sperm membranes [10]. This system contains both enzymatic and nonenzymatic antioxidants that eliminate free radicals [9]. Enzymatic antioxidants include the enzymes catalase, superoxide dismutase, glutathione reductase, and glutathione peroxidase [11]. The latter enzyme is considered to be the main enzyme responsible for the elimination of H_2_O_2_ [7]. Nonenzymatic antioxidants include substances such as urate, ascorbic acid, vitamin E, taurine, hypotaurine, carotenoids, ubiquinones, and reduced glutathione (GSH) [11]. However, the effectiveness of this system is compromised by the cryopreservation process, as it leads to changes in the activity and distribution of the supporting enzymes [10]. Oxidative stress has been associated with male infertility and low fertility rates in assisted reproductive techniques (ART). The addition of antioxidant substances to semen-freezing extenders seems to be a beneficial strategy for long-term storage, reducing the negative effects of ROS [12,13,14]. Antioxidants help overcome infertility related to oxidative stress and assisted reproduction drawbacks.

GSH is part of the largest category of antioxidants present in the body, thiols [15]. It is the main endogenous nonenzymatic antioxidant and is widely distributed in mammalian cells [16,17]. It plays an important role in defending cells against oxidative stress, including in sperm and seminal plasma [18]. In addition to being able to react with ROS, it is also used by glutathione peroxidase to reduce H_2_O_2_ to H_2_O and lipoperoxides to alkyl alcohols [13,15]. Furthermore, its protective effect is thought to be due to its ability to increase the levels of antioxidant enzymes, particularly superoxide dismutase and glutathione peroxidase, in sperm [18]. Moreover, GSH can promote a redox balance in mitochondria, preventing oxidative damage that impairs their normal function. This enables the maintenance of sperm functions, namely, sperm motility and membrane and DNA integrity, supported by mitochondrial activity [19]. Another possibility, as described in Shi et al. [19], is that GSH may reduce the decrease in GLUT 3 and 8 in ram SPZ. The maintenance of these proteins enhances the SPZ energy metabolism required for motility, viability, and membrane integrity. In this regard, several studies have described the protective role of GSH during storage and sperm cryopreservation in humans [17], equines [20,21], dogs [22], pigs [23], cattle [8,24], goats [13,18], and sheep [19].

Endogenous melatonin (MLT) participates in the control of the circadian rhythm and regulation of reproductive seasons. It also has anti-inflammatory and antioxidant properties [25]. Several studies in different species, such as humans [26], bulls [27], chickens [28], rams [12], rabbits [29], dogs [30], and rats [31], have reported that the addition of MLT to semen cryopreservation extenders has a protective effect against oxidative stress, improving sperm quality, survival, and fertility [5]. This protective effect results from its direct action, eliminating free radicals, and indirect action, stimulating the regulation of antioxidant enzyme synthesis, such as glutathione peroxidase, glutathione reductase, catalase, and superoxide dismutase [25,32,33]. MLT’s direct action on ROS during cryopreservation safeguards membrane integrity, improves mitochondrial function, increases endogenous antioxidant defenses through its influence on enzyme activity, inhibits apoptosis, and protects sperm DNA [34].

Some studies [8,35] have provided preliminary evidence of the positive effects of antioxidants on sperm physiology and fertility in humans and domestic animals by evaluating the effects on DNA integrity, sperm motility, the lipid peroxidation of sperm membranes, sperm-oocyte fusion, sperm capacitation, cAMP metabolism, and the reduction in bovine sperm motility under H_2_O_2_-induced oxidative stress. One of the objectives of this manuscript was to investigate, in native Portuguese breed rams, the effects of adding the antioxidants GSH and MLT and their combination to the semen extender on the parameters of kinetic motility and the number of live and normal SPZ (in both diluted and thawed semen).

To the best of our knowledge, there are no previous studies on the simultaneous use of GSH and MLT as antioxidant additives in ram sperm cryopreservation extenders. Considering that both substances are involved in the previously described antioxidant system, our aim was to evaluate whether the addition of these substances to the egg-yolk-based semen extender has a positive effect on the in vitro parameters of thawed ram SPZ.

## 2. Materials and Methods

### 2.1. Local and Animals

Three Portuguese Merino rams, located at the experimental farm of the Estação Zootécnica Nacional (39°11′57.008″ N 8°44′22.495″ W), were used to collect ejaculates from the 18th October to 6th December 2022. The management of these animals (between 2 and 3 years old) was similar to that previously reported by Barbas et al. [36], maintaining a body condition score of 3.25 (5-point scale) [37] and a healthy status.

These rams are from our institution and were managed in compliance with Portuguese and European legislation on animal welfare issues. This research was approved by the Institutional Review Board of INIAV protocol code 1/UEBRG/INIAV/2023.

### 2.2. Semen Extender, Antioxidant Supplementation and Experimental Design

Semen extender was formulated in the EZN laboratory according to Fernandes et al. [38] and consisted of egg yolk (15%), glycerol (6%), TRIS (2.1805 g), citric acid (1.1194 g), glucose (0.3 g), penicillin (0.05 g), and ultrapure water (Milli Q, 38 ml). We used egg yolks from fresh hen eggs, which were removed using an egg yolk separator and filter paper. We used a sterile syringe to remove the chalaza.

GSH and MLT USP powders, purchased from Sigma-Aldrich, Co. (Sintra, Portugal), were weighed and added to the previously formulated semen extender in the corresponding quantities to obtain concentrations of 5 mM of GSH and 1 mM of MLT. These concentrations were selected based on previous studies [13,39,40,41,42].

Four groups were made to assess in vitro sperm parameters: G1—semen extender without antioxidant (control group), G2—supplementation of MLT (1 mM), G3—supplementation of GSH (5 mM), and G4—supplementation of GSH (5 mM) + MLT (1 mM). After supplementing the antioxidant(s) in the respective groups, the tubes were shaken using a vortex shaker and then kept in a water bath at 30 °C until semen dilution.

Three Merino rams were used, and one ejaculate per ram was collected at each session. Eight semen collection sessions were performed for a total of twenty-four ejaculates. Sessions were held weekly on the 18th and 25th of October, from the 3rd to 29th of November, and on the 6th of December.

### 2.3. Semen Collection and Processing

Following collection with an artificial vagina, ejaculate samples were placed in a 30 °C water bath and processed and evaluated according to Fernandes et al. [38]. Only ejaculates with a volume >0.4 mL and >55% of individual motility (IM; 200× magnification at 38 °C) were processed. The ejaculates from the three rams were mixed to form a pool from each collection session. This semen pool was diluted (1:400) in a plastic cuvette with 10 μL of pure semen and 3990 μL of bidistilled water. Sperm concentration was determined by spectrophotometric analysis (WPA-S106 calibrated for sheep species). Four aliquots of equal volume with a final concentration of 800 million SPZ/mL were obtained and evaluated.

Then, 0.25 mL of diluted semen was stored with negative pressure in straws sealed with polyvinyl alcohol (PVA). Straws were placed in a water bath at 28 °C in a dry beaker and cooled to 4 °C in a refrigerated chamber for 4 h. The straws were then positioned horizontally, 4 cm higher than the liquid nitrogen, in a floating freeze rack^®^ (Minitübe GmbH, Tiefenbach, Germany), frozen in liquid nitrogen vapors (−120 °C) for 20 min, and finally submerged and stored in liquid nitrogen (−196 °C).

After a storage period of 48 h, straws were thawed in warm water (38 °C) for 1 min. The semen was homogenized with 1 mL of isotonic saline solution at 38 °C using a vortex for 3 s. Seminal evaluation was initiated two minutes later.

### 2.4. Seminal Assessment

IM, viability, and sperm morphology parameters were subjectively determined [38]. After a 1/100 dilution (10 μL of semen in 1 ml isotonic saline solution at 30 °C), 10 μL of semen was observed under a phase-contrast microscope (magnification of 200×) and platinum-heated at 37 °C. The progressive, rectilinear, and uniform movements observed in several microscopic fields were used to obtain the IM value (%).

Sperm viability (% live SPZ) and sperm morphology were determined by performing an eosin-nigrosine smear using an optic microscope (Olympus BX40 microscopic^®^, Tokyo, Japan; 1000× magnification). Normal and abnormal SPZ (%), including head, midpiece, and tail defects, were accounted. At least 100 SPZ were assessed to determine these parameters.

Kinetics and sperm trajectory movements were evaluated using the Computer-Assisted Semen Analysis system (CASA system; ISASv1^®^, Valencia, Spain) with standard settings for sheep species, i.e., head size (15–70 µm^2^) and progressive motility (80% of rectilinearity (STR)).

Our CASA system (Integrated Sperm Analysis System v1 (ISAS)^®^ [43]) has several options for semen motility evaluation. Only specific blades may be used to evaluate semen concentration. Previously, we compared sperm concentration with spectrophotometric analysis (WPA-S106) and CASA-specific blades, which was not found to be different to the results obtained with the methods presented here. Therefore, this procedure is easier, faster, and less expensive.

Total motility (TM; %), total progressive motility (TPM; %), total static (TS; %), motility subpopulations (slow, medium, and fast; %), curvilinear velocity (VCL; μm/s), linear velocity (VSL; μm/s), mean path velocity (VAP; μm/s), linearity (LIN = [VSL/VCL] × 100; %), amplitude of lateral head displacement (ALH; μm), straightness (STR = [VSL/VAP] × 100; %), head oscillation (WOB = [VAP/VCL] × 100; %) and the cross-frequency of the SPZ tail beat (BCF; Hz) were determined. The results are expressed as the average obtained by the video capture (50 images/sec) of 5 microscopic fields, with approximately 200 SPZ per field, for each group under study at each session.

All evaluated parameters were determined in diluted and thawed semen of the four groups. These methodologies are adequate, complete, and accurate and are used by several research groups for semen processing and evaluation in diluted and thawed semen [44,45].

### 2.5. Statistical Analysis

After Shapiro–Wilk W test evaluation, an arcsine square root transformation (Sqrt) of SPZ data was made to reach or approach a normal distribution.

A multivariable mixed linear model for repeated measures, using the restricted maximum likelihood (REML) method, was built for each SPZ parameter according to the following equation:Y_ijm_ = H_i_ + L_j_ + (HxL)_ij_ + t_mi_ + e_ijm_(1)
where:

Y_ijm_ is a vector of all observations and represented by the least square value;

H_i_ is the fixed effect for group (four levels: G1, G2, G3, and G4);

L_j_ is the fixed effect for processing semen (two levels: fresh sperm, thawed sperm);

(HxL)_ij_ is the two-way interaction;

T_mi_ is the random effect for session (m); 

e_ijm_ is a vector of residuals.

Differences in the fixed factors and estimate variance component of the mixed factor were evaluated using the Tukey test and Wald, respectively.

The software JMP^®^ 16 for Windows (SAS Institute, Cary, NC, USA) was used to build the models. All the results are presented as least squares means ± (Sqrt)SEM for a 0.05 level of significance.

## 3. Results

According to Table 1, all seminal parameters were negatively affected by the cryopreservation process except for intermediate piece (IP) defects. Regarding CASA sperm evaluation, an effect of the group was observed for TS, TPM, slow, medium, VCL, VSL, VAP, STR, WOB and BCF semen parameters. Semen processing × group (SP × G) interactions were observed for TPM, medium, VCL, VSL, VAP, LIN and BCF semen parameters.

Regarding the subjective evaluation of SPZ parameters, only sperm viability (dead and alive SPZ) and sperm IM were consistently affected by the freezing–thawing cycle in all groups (Table 2). Overall, abnormal SPZ remained similar between groups and freezing–thawing cycles, according to Tukey test. Only a decrease in tail defects % and an increase in head defects % after thawing were observed in G3 and G4, respectively.

The impact of semen processing was observed in TS, increasing from 2.7% in fresh semen to 38.9% in thawed semen, without differences between groups (Table 3). Conversely, TM decreased from 95.4% (fresh semen) to 58.2% (thawed semen), and no group effect was observed. In fresh semen, supplementation with 5 mM GSH (G3) had a deleterious effect on several SPZ kinetics parameters (TPM, slow, VCL, VSL, VAP, LIN, and STR) relative to the control group (G1). Supplementation with 1 mM MLT (G2) had negative effects on three SPZ parameters (TPM, VSL, and STR) compared to the control group. In G4, this adverse effect was observed for TPM, VSL, LIN, and STR compared to the control group.

In thawed semen, the highest TPM values were observed in G2, with no differences among the other groups. VSL and BCF were lower in G3 than in G2. No other differences were detected among groups for the remaining parameters. All the results are reported in Table 3.

## 4. Discussion

In this study, we observed a highly significant decrease in IM, TPM, rapid and live sperm, and a significant increase in TS (CASA) in thawed semen. However, the addition of MLT at 1 mM (G2) and GSH at 5 mM (G3), either alone or in combination (G4), did not improve SPZ parameters, with only 40% and 60% being found for IM and TM (CASA), respectively. The use of antioxidants such as MLT and GSH to prevent oxidative stress during semen storage has been studied in different species but with contradictory results [8,44]. Some authors reported that the addition of melatonin [12,46] and GSH [13,18,19] improves semen quality and motility, while others did not [47,48]. Currently, there are insufficient data to establish a correlation between the redox status and the quality of spermatozoa and embryo quality/development [8].

Cryopreservation causes damage to cellular structures, including the plasma membrane, acrosome, axoneme, and mitochondria [49,50]. These forms of damage have a direct impact on sperm motility, which relies on energy production through oxidative phosphorylation in the mitochondria and glycolysis in the fibrous sheath [51]. Therefore, if these structures are damaged during cryopreservation, ATP production is reduced, leading to impaired sperm motility [52].

Sperm motility is a critical characteristic for evaluating semen quality, as it determines the SPZ’s ability to navigate the female reproductive tract and reach the oocyte for fertilization [52]. Studies have shown that adding 100 and 200 mM GSH to fresh semen improves SPZ kinetic parameters during refrigerated storage in rams [19]. Similar effects were observed in thawed semen from bucks [13] and boars [23] when 1 and 5 mM GSH were added. However, in our study, the addition of 5 mM GSH to fresh semen did not improve sperm motility. In fact, it negatively affected (*p* < 0.05) kinetic parameters such as TPM, VCL, VSL, VAP, LIN, and STR compared to the control group. These results apparently contradict the findings of Shi et al. [19], who reported no differences in the motility parameters of ram semen immediately after GSH addition (at 0 h) but observed improvements in TPM, VCL, VSL, and VAP at 72 h of storage with 100 and 200 mM GSH.

Ansari et al. [53], Salmani et al. [48], and Banday et al. [47] also reported no beneficial effects or even detrimental effects on sperm quality parameters when GSH was added at concentrations of 3, 5, and 10 mM. These authors reported that the addition of 3 mM did not affect the TPM of buffalo semen [53] and that 5 mM affected the motility and viability of ram semen [47]. Additionally, using 5 mM GSH in buck semen did not influence TM or TPM. However, when a concentration of 10 mM was used, there was an adverse effect on the TM, VAP, and VCL parameters [48]. Considering that sperm motility patterns can be altered by the physicochemical properties of the extender, the addition of GSH before cryopreservation may have altered its inherent characteristics, impairing motility parameters [53].

The adverse effects may be attributed to osmotic stress caused by excessive GSH concentrations, as observed in ram semen [42] and equine semen [21]. Since osmolarity was not controlled in our study, negative results may have been associated with the added concentration of 5 mM GSH. Moreover, high concentrations of GSH could induce damage to the mitochondrial and axonemal structures of spermatozoa, further impairing motility and other kinetic parameters [48].

In thawed semen, there were no differences (*p* > 0.05), except for TPM, between the control and treatment groups in our study. Adding 1 mM MLT to the extender increased TPM in thawed semen, consistent with previous studies in rams [39,54]. Additionally, Fadl et al. [29] studied the effect of MLT addition on the quality and DNA integrity of thawed rabbit semen and found that supplementation with 1 mM MLT significantly increased TM and TPM compared to the control group. In addition, Shahandeh et al. [31] also demonstrated that adding 1 mM MLT to the semen extender increased TM and showed the highest TPM value in thawed rat semen.

According to Pool et al. [55], MLT can alter the functionality of SPZ subjected to cryopreservation. They suggest that MLT leads to direct action, reducing mitochondrial superoxide radical synthesis and preventing the occurrence of oxidative damage, which alters its function. It is in the mitochondria that the energy necessary for the maintenance of sperm functions is produced, namely, motility [51]. Therefore, if the addition of MLT safeguards the functionality of mitochondria, it will also allow for the maintenance of sperm motility after thawing, which may explain the increase in TPM observed in our study.

Regarding the remaining parameters, no differences were observed between groups in our study. This finding is consistent with ChaithraShree et al. [46], who reported no significant effects on sperm motility and structural changes in bovine SPZ after 48 h of cryopreservation and found that the addition of MLT had no significant effects on motility parameters, which was also verified in rat semen [31]. It should be noted that it is difficult to compare our findings with the results of other authors since most studies do not describe the CASA system configurations that were applied in their semen evaluations. This detail is very important, since the semen parameters estimated by CASA are conditionated by the selected configurations, as well as by the type of slide/camera used, the extender composition, semen processing method, and, ultimately, the operator’s choices of microscopic field for analysis [45,56].

The CASA system provides detailed values of motility and sperm kinetic parameters. TM and TPM are crucial parameters for predicting fertility, as semen needs to quickly traverse the female reproductive system to reach the oviduct for fertilization. Fast and progressive movements are particularly relevant due to the short half-life of sperm in the female genital tract, which is approximately 35–40 h. Modern CASA systems can also evaluate sperm morphology, concentration, DNA integrity, plasma membrane integrity, and vitality. However, accurately predicting semen fertility is challenging, due to the influence of multiple male and female factors. Nonetheless, semen motility, DNA integrity, and acrosome and plasma membrane integrity are frequently correlated with semen fertility [44,45].

In vitro studies provide information on the effect of antioxidants on sperm quality. Nonetheless, in vivo fertility rates following artificial insemination (AI) can be used to evaluate its efficacy and suitability in field conditions. Field studies through the AI of females provide fundamental data for the selection of the best antioxidant treatment [57]. In fact, fertility rates in flocks can be increased by supplementing extenders with antioxidants [2]. However, the results obtained regarding in vitro semen quality and in vivo fertility rate are not always consistent. The success of AI depends not only on semen quality, but also on a number of factors related to insemination technique and ewes, especially due to their reproductive anatomy [2,47].

Luo et al. [58] investigated the effects of MLT, GSH, Vit. E, and their combinations on Mediterranean buffalo semen cryopreservation. In their study, the single addition of 0.2 mM GSH or the inclusion of 1 mM MLT + 0.1 mM GSH significantly increased the viability and kinetic parameters of SPZ compared to the control group. In our study, the antioxidant combination had no beneficial effect compared to the control group. These contradictory results may be attributed to the differences among species, semen extenders, semen processing, and cryopreservation techniques, as well as the different concentrations of antioxidants and their respective combinations. Even though they are similar for MLT, the GSH concentrations were 25 (0.2 mM) to 50 (0.1 mM) times lower than in the present study. Interestingly, in our study, the addition of MLT to GSH (G4) revealed a protective effect (*p* < 0.05) because a positive effect was observed in TPM, VSL, and BCF parameters compared to the GSH group. Additionally, the group x freezing interactions confirmed these observations. This effect was similar to that reported by Alagbonsi and Olayaki [59], who investigated the MLT’s effects on the Δ9-tetrahydrocannabinol (THC)-induced reduction in rat sperm motility. They observed that, as in our study, the single inclusion of melatonin increased TPM’s and sperm’s kinetic parameters. However, when combined with THC, MLT attenuated the reduction in TPM (by 42%) and kinetic parameters caused by THC.

As previously proposed, osmotic stress may be at the origin of the adverse effects observed in the present study when using a high concentration of GSH (5 mM). Burnaugh et al. [60] found an interaction between osmotic stress and oxidative stress. These authors observed that exposure to hypo- and hyperosmotic stress in equine semen can increase the synthesis of superoxide anions by viable SPZ and consequently increase ROS formation, amplifying oxidative stress. MLT protects SPZ against oxidative stress, not only by eliminating excess ROS but also by stimulating endogenous antioxidant enzymes [34]. Several studies have described the strong antioxidant capacity of MLT. Luo et al. [58] suggested that 1 mM MLT maintains the antioxidant capacity of thawed Mediterranean buffalo sperm. Bhalothia et al. [40] showed that the addition of 1 mM MLT to ram semen extender protects SPZ by preventing free radical formation during storage at 4 °C. These facts may thus explain the protective role of MLT that is observed when MLT is combined with damaging GSH concentrations.

The detection of ROS in sperm is one of the possibilities offered by flow cytometry. This is a powerful method that is increasingly used in sperm evaluation. It can be used to easily assess and obtain information about sperm characteristics that affect semen quality and reproductive outcomes. Different probes can be used to assess ROS and changes in the oxidative status of ram sperm supplemented with different antioxidants. Some probes are specific and react with a particular anion (e.g., superoxide anion), and others are nonspecific and react with several types of ROS [61]. Fluorescent probes are employed to evaluate sperm capacitation, acrosome and plasma membrane integrity. Flow cytometry is used to accomplish a complete and accurate evaluation of thousands of sperm cells in a short time. Further in vitro studies using this methodology [55,57] are required to elucidate the effect of this antioxidant combination on the oxidative status of ram sperm. 

## 5. Conclusions

Based on the findings of our study, it can be concluded that the addition of GSH at a concentration of 5 mM did not improve, or even impaired, some kinetic parameters in fresh semen, while no differences were observed in thawed semen between the GSH (G3) and control groups. On the other hand, the addition of MLT (G2) played a protective role against the negative effects of GSH when combined (G4) in the extender. Further studies are needed to optimize the doses and combinations of these antioxidants, as well as to understand the mechanisms by which they can improve ram semen freezability. Our studies should be completed by in vitro and in vivo studies to assess the actual potential of including antioxidants in ram semen extenders.

## Figures and Tables

**Table 1 vetsci-10-00446-t001:** The effects of semen processing and treatment groups on sperm quality and motility parameters.

Parameters	Fixed Variables		Interactions
	Semen Processing (SP)	Group (G)	SP × G
**IM**	***	NS	NS
**Alive**	***	NS	NS
**Death**	***	NS	NS
**Abnormal**	**	NS	NS
**Head defects**	***	NS	NS
**IP defects**	NS	NS	NS
**Tail defects**	***	NS	NS
**TS**	***	*	NS
**TM**	***	NS	NS
**TPM**	***	**	***
**Slow**	***	*	NS
**Medium**	***	*	*
**Rapid**	***	NS	NS
**VCL**	***	*	*
**VSL**	***	***	***
**VAP**	***	***	**
**ALH**	***	NS	NS
**LIN**	**	NS	**
**STR**	***	***	NS
**WOB**	***	**	NS
**BCF**	***	*	*

*—*p* < 0.05; **—*p* < 0.01, ***—*p* < 0.001; NS—not significant (*p* > 0.05). IM—individual motility; IP—intermediate piece; TS—total static; TM—total motility; TPM—total progressive motility; VCL—curvilinear velocity; VSL—straight-line velocity; VAP—average-path velocity; ALH—amplitude of lateral head displacement; LIN—linearity; STR—straightness; WOB—wobble; BCF—beat cross-frequency.

**Table 2 vetsci-10-00446-t002:** Effects of the antioxidant’s melatonin (1 mM) and GSH (5 mM) according to subjective evaluation in fresh and thawed semen.

Parameters	G1 (Control)		G2 (Melatonin)		G3 (GSH)		G4 (GSH + Melatonin)		±(Sqrt)SEM
	Fresh	Thawed	Fresh	Thawed	Fresh	Thawed	Fresh	Thawed	
**Alive (%)**	78.7 ^a^	40.1 ^b^	79.1 ^a^	38.1 ^b^	80.9 ^a^	40.9 ^b^	82.9 ^a^	41.1 ^b^	0.17
**Death (%)**	20.9 ^a^	58.8 ^b^	20.0 ^a^	61.7 ^b^	18.6 ^a^	58.8 ^b^	16.6 ^a^	58.4 ^b^	0.22
**IM (%)**	60.0 ^a^	39.7 ^b^	58.7 ^a^	37.2 ^b^	58.7 ^a^	40.3 ^b^	59.3 ^a^	38.0 ^b^	0.08
**Abnormal (%)**	15.3 ^a^	11.6 ^a^	15.5 ^a^	11.4 ^a^	20.3 ^a^	10.8 ^a^	15.0 ^a^	12.9 ^a^	0.35
**Head D (%)**	0.8 ^a,b^	2.6 ^a,b^	1.2 ^a,b^	1.7 ^a,b^	1.1 ^a,b^	1.7 ^a,b^	0.7 ^b^	3.0 ^a^	0.37
**IP D (%)**	3.1 ^a^	3.8 ^a^	1.9 ^a^	2.7 ^a^	2.2 ^a^	2.2 ^a^	1.5 ^a^	2.1 ^a^	0.24
**Tail D (%)**	9.6 ^a,b,c^	3.8 ^c^	10.9 ^a,b^	5.3 ^b,c^	15.3 ^a^	5.5 ^b,c^	11.0 ^a,b^	6.3 ^b,c^	0.40

^a–c^ different superscript letters in the same row: *p* < 0.05. IM—individual motility; IP—intermediate piece; D—defects; GSH—reduced glutathione.

**Table 3 vetsci-10-00446-t003:** Effects of the antioxidants melatonin (1 mM) and GSH (5 mM) on the motility and kinetic parameters of spermatozoa evaluated by CASA.

Parameters	G1 (Control)		G2 (Melatonin)		G3 (GSH)		G4 (GSH + Melatonin)		±(Sqrt)SEM
	Fresh	Thawed	Fresh	Thawed	Fresh	Thawed	Fresh	Thawed	
TS	4.1 ^a^	37.6 ^b^	2.7 ^a^	35.3 ^b^	4.4 ^a^	38.9 ^b^	2.9 ^a^	36.8 ^b^	0.17
TM	93.4 ^a^	59.5 ^b^	95.4 ^a^	62.3 ^b^	94.0 ^a^	58.2 ^b^	95.4 ^a^	61.1 ^b^	0.11
TPM	49.2 ^a^	20.5 ^b^	40.5 ^c^	26.7 ^d^	39.7 ^c^	20.8 ^b^	37.2 ^c^	23.6 ^b,d^	0.21
Slow	0.5 ^a^	3.4 ^b^	0.4 ^a^	3.2 ^b^	2.3 ^b^	3.7 ^b^	0.4 ^a^	4.5 ^b^	0.23
Medium	3.5 ^b,c^	6.1 ^a^	3.5 ^b,c^	5.5 ^a,b^	5.8 ^a,b^	5.9 ^a,b^	2.6 ^c^	5.8 ^a,b^	0.20
Rapid	87.1 ^a^	46.9 ^b^	89.0 ^a^	50.3 ^b^	83.0 ^a^	44.0 ^b^	89.5 ^a,b^	44.8 ^b^	0.21
VCL	222.0 ^a^	137.2 ^c^	211.3 ^a,b^	147.5 ^c^	197.8 ^b^	136.3 ^c^	216.6 ^a,b^	135.8 ^c^	0.29
VSL	85.8 ^a^	49.5 ^c,d^	75.3 ^b^	56.6 ^c^	69.0 ^b^	49.0 ^d^	75.0 ^b^	52.5 ^c,d^	0.19
VAP	110.6 ^a^	70.9 ^c^	104.5 ^a,b^	76.3 ^c^	94.6 ^b^	68.9 ^c^	107.3 ^a^	71.0 ^c^	0.20
ALH	3.6 ^a^	2.9 ^b^	3.6 ^a^	2.9 ^b^	3.5 ^a^	2.9 ^b^	3.6 ^a^	2.8 ^b^	0.03
LIN	38.2 ^a^	36.0 ^a,b^	35.4 ^a,b^	38.3 ^a^	34.5 ^b^	35.9 ^a,b^	34.7 ^b^	38.5 ^a^	0.09
STR	77.2 ^a^	69.7 ^b,c^	71.5 ^b,c^	74.1 ^a,b^	72.5 ^b,c^	70.8 ^b,c^	69.6 ^c^	73.3 ^a,b,c^	0.09
WOB	49.4 ^b,c,d^	51.3 ^a,b,c^	49.2 ^c,d^	51.5 ^a,b^	47.5 ^d^	50.4 ^a,b,c^	49.4 ^b,c,d^	52.2 ^a^	0.05
BCF	22.8 ^a^	18.5 ^c,d^	20.8 ^a,b^	19.7 ^b,c^	20.8 ^a,b^	17.6 ^d^	21.4 ^a,b^	19.2 ^b,c,d^	0.09

^a–d^ different superscript letters in the same row: *p* < 0.05. GSH—reduced glutathione; TS—total static (%); TM—total motility (%); TPM—total progressive motility (%); slow (%); medium (%); rapid (%); VCL—curvilinear velocity (μm/s); VSL—straight-line velocity (μm/s); VAP—average-path velocity (μm/s); ALH—amplitude of lateral head displacement (μm); LIN—linearity (%); STR—straightness (%); WOB—wobble (%); BCF—beat cross-frequency (Hz).

## Data Availability

The data presented in this study can be obtained upon request from the corresponding author.

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
