# Peer review of "Effect of In Vitro Addition of Melatonin and Glutathione on Seminal Parameters of Rams in Diluted Semen and after Thawing"

_vetsci, 2023, doi:10.3390/vetsci10070446_

Round 1

Reviewer 1 Report

The authors could give more information about their choice to use a glass slide covered with a coverslip for the evaluation of kinematic parameters of the diluted semen and not specific slides for CASA systems (pp 187-188, materials and methods)

Author Response

The authors could give more information about their choice to use a glass slide covered with a coverslip for the evaluation of kinematic parameters of the diluted semen and not specific slides for CASA systems (pp, materials and methods).

L187-192: “Our CASA system (Integrated Sperm Analysis System v1 (ISAS)® [39]) have in its settings several options for semen motility evaluation. Only specific blades may be used for evaluation semen concentration. Previouly we compared  sperm concentration with spectrophotometric analysis (WPA-S106) and with CASA specific blades, which didn´t show differences with our current methods. So, this procedure seasier, faster and less expensive.”

Reviewer 2 Report

Comments to the manuscript ID: vetsci-2445217

The present manuscript describes the effect of melatonin and glutathione on ram semen quality after thawing. The topic covered in this work is very important.

However, some recommendations and questions must be addressed before its publication in Veterinary Sciences journal

I understand the untiring efforts made by the authors; however, some aspects of the manuscript need attention of the authors for improving the quality of manuscript. The authors need to address below queries for improving the manuscript content.

General Comments

The written quality of the Manuscript should be much improved. The Manuscript must be proofread by a native English speaker or professional editing service. Not only the grammar and spelling, but also the narrative needs improvements; a better flow throughout the text is required.

No ethical permission has been cited in the manuscript for conducting the study.

The methodology used is very poor. The authors need to add more test such as flow cytometry ….

What is the novelty of your work?

The study lacks validation of outcomes as no in vitro or in vivo fertility trials have been carried out in this study.

in my opinion the authors can present only the results after thawing

Specific Comments

Title: the title should be reformulated. Because the title must resume the content of the work. you presented a work on fresh and cryopreseravtion

Abstract.

Line 33-36: please change the fresh by diluted or refrigerated semen.

Line 38-40: How about cryopreservation??

Introduction

Line 76-77. In this regard, several studies. please add more reference because the 18 is only for ram .

please provide some hypothesis about the protective effect of antioxidant

Materials and Methods

Please add Ethical statements or ethical approval

The authors need to add more test such as flow cytometry...

Please verify the statistical analysis. The authors showed compared the difference fresh in the groups ; difference in cryopreservation groups and fresh vs cryopres

Example: table 3

20.5 b 26.720.8 b 23.6 b,d were is a?

Figure 1. is not necessary

Results

Please Rewrite the results part, because there is some confusion.

Discussion

Please avoid the general stated and the redundancy because you need the discuses your results.

The written quality of the Manuscript should be much improved. The Manuscript must be proofread by a native English speaker or professional editing service. Not only the grammar and spelling, but also the narrative needs improvements; a better flow throughout the text is required.

Author Response

Comments to the manuscript ID: vetsci-2445217

The present manuscript describes the effect of melatonin and glutathione on ram semen quality after thawing. The topic covered in this work is very important.

However, some recommendations and questions must be addressed before its publication in Veterinary Sciences journal

I understand the untiring efforts made by the authors; however, some aspects of the manuscript need attention of the authors for improving the quality of manuscript. The authors need to address below queries for improving the manuscript content.

General Comments

The written quality of the Manuscript should be much improved. The Manuscript must be proofread by a native English speaker or professional editing service. Not only the grammar and spelling, but also the narrative needs improvements; a better flow throughout the text is required.

No ethical permission has been cited in the manuscript for conducting the study.

L130-133: These rams are from our institution and were managed in accordance with Portuguese and European legislation on animal welfare issues. This research was approved by the Institutional Review Board of INIAV protocol code 1/UEBRG/INIAV/2023.

The methodology used is very poor. The authors need to add more test such as flow cytometry ….

Only CASA system was used in the present study. We expect to use flow cytometry in a near future.

What is the novelty of your work?

L118-119: “To our best knowledge, there are no previous studies on the concomitant use of GSH and melatonin as antioxidant supplements added to extenders for cryopreservation of ram sperm.”

The study lacks validation of outcomes as no in vitro or in vivo fertility trials have been carried out in this study.

This is an in vitro study presenting findings related to the effect of antioxidants in sperm traits, which is necessary to perform in vitro and in vivo fertility trials

in my opinion the authors can present only the results after thawing

This can be an option, but in the present study we evaluated the effect of processing semen and groups as a whole model (i.e., we can compare 8 situations at same time); this allows to compare values before and after thawing and respective interactions. The fresh semen is the “raw material” which value of the sperm traits also depend.

Specific Comments

Title: the title should be reformulated. Because the title must resume the content of the work. you presented a work on fresh and cryopreseravtion

L3: “…parameters of rams in diluted semen and after thawing…”

Abstract.

Line 33-36: please change the fresh by diluted or refrigerated semen.

Done.

Line 38-40: How about cryopreservation??

L36-39: In thawed semen, 1 mM of melatonin increased the TPM regarding the group control. VSL was lower in GSH 5mM group than in MLT 1mM group. Also, the combination of both antioxidants attenuates the negative effect of GSH 5mM on TPM, VSL and BCF.”.

L40: “…does not improve sperm kinetic parameters in both fresh and thawed sémen…”

Introduction

Line 76-77. In this regard, several studies. please add more reference because the 18 is only for ram .

L93-94: “… both in humans [17], equines [20,21], dogs [22], pigs [23], cattle [8,24], goats [13,18], and sheep [19].”.

please provide some hypothesis about the protective effect of antioxidant.

L83-92: “In addition, its protective effect is thought to be due to its ability to increase the levels of antioxidant enzymes, particularly superoxide dismutase and glutathione peroxidase in sperm [18]. Moreover, GSH can promote redox balance in mitochondria, preventing oxidative damage that impairs their normal function. This enables the maintenance of sperm functions supported by mitochondrial activity, namely sperm motility and membrane and DNA integrity [19]. Another possibility described by Shi et al. [19] is that GSH may reduce the decrease in the amount of GLUT 3 and 8 in ram spermatozoa. Maintenance of these proteins enhances energy metabolism and allows sperm to be supplied with energy required to maintain sperm motility, viability, and membrane integrity”

Materials and Methods

Please add Ethical statements or ethical approval

Done. See above.

The authors need to add more test such as flow cytometry...

See previous comment.

Please verify the statistical analysis. The authors showed compared the difference fresh in the groups; difference in cryopreservation groups and fresh vs cryopres

Example: table 3

20.5 b 26.720.8 b 23.6 b,d were is a?

The model used evaluate the effect of the semen processing, semen group and their interactions at same time. As consequence, the comparison between groups is done considering 8 groups, i.e., all the superscript letters in the same row should be taken in consideration, e.g.:

TPM

49.2 a

20.5 b

40.5 c

26.7 d

39.7 c

20.8 b

37.2 c

23.6 b,d

 Nevertheless, we confirmed the statistic, and corrected the significance of effects for STR in Table 1. Thanks.

Figure 1. is not necessary

The figure 1 is in fact a graphical presentation of the Table 3, and it is not essential. It was removed.

Results

Please Rewrite the results part, because there is some confusion.

This part was improved presenting the main values in the text.

Discussion

Please avoid the general stated and the redundancy because you need the discuses your results.

We moved the first paragraph below the first description of our results and added a paragraph about the observed results for melatonin (L318-326). We believe that can help to better understand our results.

Comments on the Quality of English Language

The written quality of the Manuscript should be much improved. The Manuscript must be proofread by a native English speaker or professional editing service. Not only the grammar and spelling, but also the narrative needs improvements; a better flow throughout the text is required.

The English language was improved.

Reviewer 3 Report

  This manuscript deals with the suitability of two antioxidants ( glutathione and melatonin) and its combination for avoiding the oxidative stress in the conservation of fresh and thawed semen from rams.  Conservation of ram semen is a very important problem for artificial insemination in sheep, where usually fresh semen is used as a consequence of the problems caused by the freezing and subsequent thawing of the semen of this species. For this reason, the subject and objectives of this manuscript are very interesting for sheep specialist veterinarians. The manuscript is well written and easy to read. The statistics methods are adequate to the nature of data.  The authors have made a great effort for resuming the numerous and varied results in tables and figures.  The discussion includes an extensive literature review. The conclusions are fully supported by the results. The reference list is updated: 25/49 ( 51%) of references are from the last 5 years. For these reasons, in the opinion of this reviewer, the manuscript deserves to be accepted for publication with the following (minor) suggestions.

 General suggestion: Parameters were measured in thawed semen, as said in the title of manuscript and in the caption of tables.  However , “frozen semen” appears many times throughout the text and at the bottom of figure 1. It may be worth substituting “frozen semen” for “thawed semen” throughout the manuscript.

 Line 159: How long did the semen remain frozen before being thawed?

 Line 224: According to Table 1, a significant interaction was also observed for BCF

Lines 344-345:   It is said: Even though they are similar for MLT, yet GSH concentration was 25 (0.2 mM) to 50 (mM). Maybe 0.1 must be added before in 50 (mM) before mM?

Author Response

 General suggestion: Parameters were measured in thawed semen, as said in the title of manuscript and in the caption of tables.  However , “frozen semen” appears many times throughout the text and at the bottom of figure 1. It may be worth substituting “frozen semen” for “thawed semen” throughout the manuscript.

Done in the whole manuscript.

 Line 159: How long did the semen remain frozen before being thawed?

L169: “After a storage period of 48h, straws were thawed...”.

 Line 224: According to Table 1, a significant interaction was also observed for BCF

L228: Corrected

Lines 344-345:   It is said: Even though they are similar for MLT, yet GSH concentration was 25 (0.2 mM) to 50 (mM). Maybe 0.1 must be added before in 50 (mM) before mM?

L345-346: Corrected

Round 2

Reviewer 2 Report

Comments to the manuscript ID: vetsci-2445217

The present manuscript describes the effect of melatonin and glutathione on ram semen quality

The authors have made a lot of improvement of their paper. However, some questions need to be addressed before the publication: 

1- What is the impact of your work on the field of Animal reproduction?

2- Why the authors choose (MLT 1mM, GSH 5mM, and MLT 1mM + GSH 5mM)?

3- Did you think that only CASA could give some prediction on the fertility results? 

4- The authors need to add flow Cytometry and ROS production to confirm the effect of Antioxidant

5- Could you try to add some IVF or AI in order to confirm your data

Comments to the manuscript ID: vetsci-2445217

The present manuscript describes the effect of melatonin and glutathione on ram semen quality.

The authors have made a lot of improvement of their paper. However, English needs some improvement, especially in the length of the sentence. 

Author Response

1- What is the impact of your work on the field of Animal reproduction?

The most important aim of this study was to evaluate effect of the antioxidant’s combination (MLT and GSH) on in vitro thawed SPZ parameters of rams (L105-106). No beneficial effect compared to the control group were observed (L334). It was suggested that MLT can be a more efficient action and can play a protective role of adverse effects when GSH is added (L339-341). This impact was highlighted in the conclusions section (362-366).

2- Why the authors choose (MLT 1mM, GSH 5mM, and MLT 1mM + GSH 5mM)?

L130: “These concentrations were selected based on previous studies [13,39–42].”

3- Did you think that only CASA could give some prediction on the fertility results? 

L320-329: “The CASA system provides detailed values of motility and sperm kinetics parameters. TM and TPM are crucial parameters for predicting fertility, as semen needs to quickly traverse the female reproductive system to reach the oviduct for fertilization. Fast and progressive movements are particularly relevant due to the short half-life of sperm in the female genital tract, which is approximately 35-40 h. Modern CASA systems can also evaluate sperm morphology, concentration, DNA integrity, plasma membrane integrity and vitality. However, accurately predicting semen fertility is challenging due to the influence of multiple male and female factors. Nonetheless, semen motility, DNA integrity, and acrosome and plasma membrane integrity are frequently correlated with semen fertility [44,45].”

4- The authors need to add flow Cytometry and ROS production to confirm the effect of Antioxidant

Both evaluations are important, but will be not possible to perform them in the present study. Please see L368-370 suggesting further studies in this topic.

5- Could you try to add some IVF or AI in order to confirm your data

For instance, we don’t have appropriate data to report these results. Please also see L368-370.

The authors have made a lot of improvement of their paper. However, English needs some improvement, especially in the length of the sentence.

The document was subjected to an improvement regarding English language. Several changes were made in the whole manuscript. Also, all abbreviations were checked and corrected when appropriate.
